# Porcine Epidemic Diarrhea Virus: An Updated Overview of Virus Epidemiology, Virulence Variation Patterns and Virus–Host Interactions

**DOI:** 10.3390/v14112434

**Published:** 2022-11-02

**Authors:** Yuanzhu Zhang, Yiwu Chen, Jian Zhou, Xi Wang, Lerong Ma, Jianing Li, Lin Yang, Hongming Yuan, Daxin Pang, Hongsheng Ouyang

**Affiliations:** 1Key Laboratory of Zoonosis Research, Ministry of Education, College of Animal Sciences, Jilin University, Changchun 130062, China; 2Chongqing Research Institute, Jilin University, Chongqing 401120, China; 3Chongqing Jitang Biotechnology Research Institute Co., Ltd., Chongqing 401120, China

**Keywords:** PED, PEDV, swine, epidemiology, virulence, virus–host interaction

## Abstract

The porcine epidemic diarrhea virus (PEDV) is a member of the coronavirus family, causing deadly watery diarrhea in newborn piglets. The global pandemic of PEDV, with significant morbidity and mortality, poses a huge threat to the swine industry. The currently developed vaccines and drugs are only effective against the classic GI strains that were prevalent before 2010, while there is no effective control against the GII variant strains that are currently a global pandemic. In this review, we summarize the latest progress in the biology of PEDV, including its transmission and origin, structure and function, evolution, and virus–host interaction, in an attempt to find the potential virulence factors influencing PEDV pathogenesis. We conclude with the mechanism by which PEDV components antagonize the immune responses of the virus, and the role of host factors in virus infection. Essentially, this review serves as a valuable reference for the development of attenuated virus vaccines and the potential of host factors as antiviral targets for the prevention and control of PEDV infection.

## 1. Epidemiology of PEDV

Porcine epidemic diarrhea (PED) is an infectious intestinal disease caused by the porcine epidemic diarrhea virus (PEDV) [1]. PEDV mainly affects piglets more severely than adult pigs, generally showing symptoms such as vomiting, liquid diarrhea, dehydration, anorexia, and weight loss [2]. The presence of PEDV was first reported on a farm in the United Kingdom in 1971 and the virus was first isolated in Belgium in 1978 [3]. In 1984, China confirmed the existence of PED antigen for the first time through the serum antigen test [1]. In early 2004, PEDV with low mortality did not receive much global attention. By 2010, high-mortality strains of PEDV emerged and spread globally, having an increasing impact on the global swine industry. In this regard, a comprehensive understanding of PEDV has become increasingly urgent. Here, we conduct a comprehensive review of the epidemiology of PEDV, including discussing the worldwide spread of PEDV variants, exploring the origins of PEDV variants around the world, and describing the structure and function of PEDV.

### 1.1. The Morbidity and Mortality of PEDV

Mortality was often associated with the age of recipient animals after PEDV infection [4]. In aged animals, although the morbidity may be close to 100%, the mortality rate is very low, between 1% and 3% [5]. However, the morbidity in piglets is more severe, with an average of 50% mortality in piglets within two weeks of birth [6,7]. In 2004, PEDV infection caused a regional outbreak with a low fatality rate. At this point, the morbidity in piglets was 46.4%, and the mortality rate was only 6.16% [1]. In the winter of 2010, a new variant of highly pathogenic GII PEDV appeared in southern China, which increased the morbidity and mortality of PEDV to more than 80–100% [8,9]. More than a million piglets, even vaccinated ones, died on pig farms in southern China, dealing a devastating blow to the swine industry [8,10]. In 2013, according to statistics, morbidity in the United States was nearly 100%, and piglet mortality was 90–95% [11,12]. As of 2014, the mortality rate of piglets infected with pathogenic strains in Germany was more than 70% [13]. From 2016 to 2018, the mortality rate of Mexican piglets was 80–95% [14]. In recent years, the most popular GII PEDV highly pathogenic variant has gradually spread around the world, and the threat to the global swine industry has become more and more serious.

### 1.2. Transmission of PEDV

Transmission by direct and indirect contact is the main route of PEDV transmission between pigs. PEDV is transmitted between pigs at different ages and usually first infects finishing pigs, then the virus spreads to pregnant sows in the farrowing house, and subclinically infected sows then transmit PEDV to suckling piglets, eventually causing an epidemic of high mortality in piglets. Here, we summarize several routes of PEDV transmission between pigs (Figure 1).

PEDV spreads in several different ways during infection, and one of the most common is the fecal–oral route, that is, direct or indirect transmission through porcine feces, vomit, and other contaminants produced by the livestock process [7]. PEDV can be infected through the route of transmission from porcine feces to the nasal cavity, known as the fecal–nasal route [15]. In addition, PEDV can also be transmitted vertically to piglets through sow milk [8]. According to a study by Gallien et al. (2018) [5], the presence of PEDV can be detected in the semen of PEDV-infected boars, and prolonged virus shedding can be observed in the sperm-rich fraction, proving that PEDV can be sexually transmitted through boar semen [5]. A recent study showed that PEDV infected by nasal epithelium can be transferred to CD3+ T lymphocytes through synapses, and finally reaches the intestinal mucosa through blood circulation, resulting in intestinal infection [16], which is the mechanism of PEDV infection from the nasal cavity to the intestinal epithelium, and is also important evidence for aerosol transmission. In addition to pig–to–pig infection, the transmission of PEDV can also occur through contact with contaminated equipment [17], contaminated vehicles used for animal transport [18], or farm employees [16,17].

### 1.3. Genotyping, Distribution, and Origin of PEDV

PEDV continuously evolves in the process of adapting to environmental changes, and its frequent variants lead to the emergence of various virulent strains. At present, researchers have classified PEDV into the GI classical genogroup and the GII variant genogroup [19]. Genogroups GI and GII further evolved separately, and finally formed five subgroups (GIa, GIb, GIIa, GIIb, and GIIc). To further clarify the genotype characteristics of PEDV and the global prevalence of each genogroup, we discuss the distribution and origin of each genogroup of PEDV in Table 1.

Genogroup GI is divided into two subgroups, GIa and GIb. The GIa subgroup mainly consists of classic CV777, DR13, SM98, and other early strains, distributed in Europe and Asia [32]. The GIb strains mainly originate from Asian countries, especially China [33], which are generally early strains or cell-adapted attenuated vaccine strains caused by the insertion or deletion of the PEDV *S* gene. Compared with the GIa with a certain virulence, the GIb strains are less virulent, and many attenuated strains have been identified as candidates for attenuated vaccines, which we propose to term “S-Indel vaccine strains”. The most typical strains in the GIb subgroup include the early strain JS-2004-2 [22] and vaccine strains, such as the attenuated CV777 [34], attenuated DR13 [24], and SD-M [35]. In addition to the typical attenuated vaccine strains, there is also the CV777 vaccine strain-like AH-M classic strain, which has been circulating in China recently [36].

Since 2010, genogroup GII strains have been predominant globally, especially in China [26]. The GI and GII strains are not genetically closely related, and one of the most typical differences is that the GII strains have 11 different amino acid mutations (I116T, I356T, E365Q, T549S, G594S, N724S, A959V, S1044A, G1173D, S1232R, and R1298Q) [19]. Subgroup GIIa is an emerging non-S INDEL strain recombined within the S protein domain with higher virulence than genogroup GI, including those prevalent in Asia (AH2012 [37], LZW [38], ZMDZY [39], GDZQ [40], and Tottori2 [41]), and those prevalent in North America (Colorado [42], PC21A [43], and PC177 [44]), which occurs more frequently in Asia [33]. The GIIb subgroup is also a highly virulent non-S INDEL strain recombined within the S protein domain. All GIIb strains are currently circulating in China, including AJ1102 [45], LC [46], AH2012-12 [37], YN1 [47], and other typical strains [27,48,49,50]. However, no potential recombination occurred in the SH strain of GIIb [30], suggesting that amino acid substitutions in the S protein are also a pattern for the GIIb subgroup. The third important subtype of GII is GIIc, a kind of S-INDEL strain produced by recombination of the subgroups GIa and GIIa and the structural changes of the *S* gene [19,22]. These strains commonly have one amino acid insertion (aa 161–162) and two amino acid deletions (aa 59–62 and aa 140) in the N-terminal region of the S protein. Compared with other subgroups, GIIc also has three unique amino acid substitutions (L76, A/S92 and H/T113) and antigenic drift due to amino acid substitutions [19]. The less virulent GIIc strains, such as OH851, 15V010, CH/HNQX-3/14, and ZL29 [31,51,52,53], are mainly disseminated and prevalent in Europe, with a minority from the United States and Asia [4].

PEDV is mainly prevalent in Asia, North America, and Europe. Exploring the global prevalence and transmission of PEDV is thus relevant to understanding the origin of PEDV in various continents. The sporadic outbreaks of PEDV in 2005 occurred only in Asia, and they all belonged to the classic GIa strains [54]. In October 2010, a highly virulent GII variant which was different from the classic strain CV777 appeared in China for the first time [10]. Since then, China [8], South Korea [55], Japan [56], Vietnam [57], Thailand [58], and the Philippines [59] have suffered an outbreak of PEDV, successively. Among them, two pandemic strains, MG868 and BIG1024, emerged in South Korea, and these are likely to be the origin of the subsequent outbreak of PEDV in Asia [1]. In April 2013, when a highly virulent non-S INDEL GII strain was detected for the first time in the United States [60,61], PEDV appeared in the United States and rapidly spread throughout North America, including Canada [62] and Mexico [63]. Here, the three representative emerging strains, MN, IA1, and IA2, were closely related to the AH2012 strain in China [61]; however, the potential parental PEDV strain from which the North American outbreak originated has not been identified. In 2014, a newly discovered mild PEDV S-INDEL strain (GIIc) emerged in Germany [64] and began to circulate in many European countries such as the Netherlands [65], France [66], Belgium [53], Portugal [67] and Austria [68]. These mild variants were speculated to be homologous with the PEDV variant strain OH851, which was first detected in the United States earlier that same year [51].

### 1.4. Virion Structure and Function of PEDV

PEDV is a single-stranded positive-sense RNA virus with a diameter of 95–190 nm, a typical nested crown, and a genome size of approximately 28 kb, belonging to the genus alphacoronavirus [69,70]. PEDV consists of seven open reading frames (ORF1a, ORF1b, and ORF2–6), of which ORF1a (nt 297–12650) and ORF1b (nt 12605–20641) encode a total of sixteen nonstructural proteins, including nsp1~nsp16; ORF2 and ORF4–6 encode four structural proteins, including the spike (S) protein of 150–220 kDa, the envelope (E) protein of 7 kDa, the membrane (M) protein of 20–30 kDa, and the nucleocapsid (N) protein of 58 kDa; ORF3 encodes an accessory protein ORF3 [4,24,70,71] (Figure 2).

According to the molecular structure of the PEDV genome drawn, the ORF1a and ORF1b regions start from the 5′-end cap, and the polyproteins ppla and pp1ab translated from its replicase gene are post-translationally cleaved by internal proteases, resulting in 16 mature end-product nonstructural proteins nsp1–16 [4,70], mainly playing a role in virus infection; this can inhibit the immune responses of the host and create a favorable environment for virus infection and proliferation [71].

The S protein is located on the outer layer of the viral envelope, a type I glycoprotein, which is the most basic functional protein in PEDV [72]. Compared with the homology analysis of the corresponding regions of other coronaviruses, the S protein of PEDV is divided into two parts: S1 (19–726 aa) and S2 (727–1383 aa) [73]. The S1 subunit contains two RBDs, S1-NTD and S1-CTD [74], which mainly bind to receptors to facilitate viral attachment [75]. The S2 subunit contains three domains, two heptapeptide repeat domains (HR1 and HR2, 978–1117 aa and 1274–1313 aa) and a transmembrane domain (TM, 1324–1346 aa) [76], which mainly promote viral membrane fusion [77]. Four B cell epitopes have been characterized in the S protein, including the core neutralizing epitope (COE) (499–638 aa) on the S1 subunit, the epitopes SS2 (748–755 aa), SS6 (764–771 aa) and 2C10 (1368–1374 aa) on the S2 subunit [78], which become the main target for the development of vaccines or antibodies. The S protein plays an important role in viral pathogenicity, tissue tropism, infection, dissemination, and the trypsin-dependent proliferation of PEDV [73,79,80,81,82].

The accessory protein ORF3 is the characteristic protein of coronavirus and the only accessory protein in PEDV. ORF3 has a homotetrameric structure comprising four transmembrane regions, which are generally encoded by sequences located between the viral S and E genes [83]. The ORF3 protein of PEDV has a similar function to that of SARS coronavirus, and the intact ORF3 protein can function as an ion channel and regulate virus release during infection [84,85]. In addition, the ORF3 gene of PEDV has been shown to prolong the S phase of host cells by promoting the formation of vesicular structures, indicating its importance in the process of PEDV replication [86]. In general, ORF3 plays an important adjunctive role in PEDV infection, especially in the interaction between the virus and host cells.

The E protein is mainly localized in the endoplasmic reticulum, with a minor amount in the nucleus [87], whose molecular structure is divided into three regions, including a short amino-terminal hydrophilic region, an alpha-helical transmembrane domain of approximately 25 aa in length, and a long carboxyl-terminal region [88]. E protein functionally does not affect host cell proliferation and cell cycle, but it is implicated in the inflammatory response and persistence of PEDV infection [87]. Besides this, the E protein can also help evade host innate immunity by inhibiting RIG-I-mediated signaling [89]. Finally, the E protein tends to induce membrane curvature, and plays an important role in virion morphogenesis, assisting in the assembly of coronaviruses [90,91,92].

The M protein is an important membrane glycoprotein within the envelope, and the most abundant component in the viral envelope, which can interact with the E and S proteins encoded upstream of the viral structure, to complete the envelope assembly of the coronavirus [93]. The M protein can also interact with ORF3, and together with the N protein, participate in the assembly and budding of virus particles [94,95]. Additionally, the M protein suppresses the host immune responses by inducing neutralizing antibodies or inhibiting interferon-β (*IFN*-β) activity [96,97], while inducing cell-cycle arrest at S-phase through the cyclin A pathway during viral infection [98].

The N protein, called the nucleocapsid protein, is the main structural protein involved in the formation of the helical nucleocapsid, together with the viral genomic RNA [99]. It is currently known that in coronaviruses, the N protein is the only phosphorylated basic structural protein that is required for the efficient replication and transcription of viral RNA, and assists in the process of organizing the viral genome to help virion assembly [99,100,101]. During viral infection, PEDV N can regulate the cell cycle and suppress the immune responses by *IFN* regulatory factor 3 (IRF3) [102]. Meanwhile, PEDV N induces endoplasmic reticulum stress and inhibits cell-induced apoptosis through interaction with cellular phosphoproteins [103,104]. Surprisingly, the PEDV N protein may help to enhance the replication of viruses closely related to PEDV, such as porcine reproductive and respiratory syndrome virus (PRRSV), whereas it is inactive against unrelated viruses [105].

## 2. Factors Affecting Pathogenicity

Compared to GIIc strains with lower mortality rates in Europe, the prevalence of highly virulent GII strains with almost 100% mortality in Asia and the Americas caused greater losses to the swine industry. Moreover, due to the high heterogeneity of PEDV, vaccines developed for GI strains cannot effectively prevent and treat infections caused by highly virulent GII strains. Comparing the heterogeneity between variant strains and classical strains and analyzing the impact of these variants on virulence changes will help to parse out important factors affecting virus virulence and assess in a timely way the potential threat of emerging variants to the swine industry. Here, we describe the effects of PEDV viral proteins on virulence change and growth adaptation of strains from the perspective of gene heterogeneity. Finally, we try to find the key amino acids on the S protein that affect virus virulence and cell growth fitness, which have important reference values for the development of attenuated vaccines.

### 2.1. Variation in Structural/Accessory/Non-Structural Proteins

The viral S protein, which is involved in receptor recognition and facilitates membrane fusion and anchoring, is a major factor in host tropism and pathogenesis of coronaviruses and may play important roles in virulence and host or tissue tropism [106,107,108]. To explore the effect of strain variation on virus virulence, in Table 2, we summarize the amino acid indels in the genome of several variant strains with the highly virulent parental strains as a reference.

The S1 subunit of the S protein is an important determinant of PEDV virulence, using reverse genetics [118]. Of these attenuated PEDV strains, most have a deletion in a stretch of amino acid sequences of the S gene, which clarifies the role of the S protein in virulence. The PEDV S protein contains two sorting motifs in the cytoplasmic tail: endocytosis signal-YxxΦ and ER retrieval signal (ERRS)-KVHVQ. Some studies have reported that deletion of the conserved motif of YxxΦEKVHVQ can prematurely truncate the S protein and reduce the pathogenicity in piglets [114], which clarifies the effect of the conserved motif of the S protein on virulence.

The S protein is not the only factor affecting viral virulence; ORF3 also plays a role in enhancing viral virulence. In contrast to the BJ2011C strain infected with a severe phenotype, the CHM2013-S_BJ_ strain with a 70–aa deletion in ORF3 did not have any pathogenic properties during infection, underscoring the role of ORF3 in conferring virulence [122]. Similarly, the attenuated DR13 strain lacks 16 amino acids in the ORF3 gene, making it a candidate vaccine against PEDV infection [123]. However, not all ORF3 truncations are effective in reducing pathogenicity; the strains P55, 17GXCZ-1ORF3d and HN2021 with field naturally truncated ORF3 were all lethally virulent compared to field isolates with intact ORF3 [122,123,124]. Of course, the field strains CHM2013, SM98, and AVCT12 in which 10 amino acids were deleted continuously from the start codon resulting in a full-length truncation of ORF3, had mild phenotypes [55,117,118]. These could be explained by the fact that in field isolates, only the complete truncated full-length ORF3 strain may contribute to the reduction of pathogenicity. It has been shown that ORF3 enhances virus production through ion channel activity [120]. A mutation at residue 170 of ORF3 (Y170A) can significantly reduce its ionic activity [84], which may suggest that the Y170A mutation can be considered one of the targets for reducing virulence.

Studies have shown that the nonstructural protein nsp16, a 2′-O-methyltransferase (2′-O-MTase), also partially attenuates the pathogenicity of severe acute respiratory syndrome coronavirus (SARS-CoV) [121], and may be another important cofactor affecting virus virulence. Studies have confirmed that the attenuated KDKE^4A^-SYA mutant was constructed by inactivating 2′-O-MTase activity with quadruple alanine substitutions (K45A-D129A-K169A-E202A) and inactivating the endocytic signal of the S protein with mutation (Y1378A), which is a new candidate for live attenuated vaccines [109].

### 2.2. Sequence Heterogeneity of S, ORF3, E, M, and N Genes

The availability of attenuated vaccine strains is a major concern, due to the extensive genetic diversity of PEDV strains. To comprehensively study the genetic diversity of PEDV, here, we describe the contribution of each viral protein to the genetic diversity of PEDV from the perspective of viral genome conservation.

The study showed that in 28 Korean field samples, M (0–4%), ORF3 (0–3.9%), and E (0–5.3%) were more conservative than S (0.1–8.4%) and N (0.1–6.8%) [55]. Another report pointed out that compared with nsps, M, and N proteins, structural proteins S, and E, and accessory proteins ORF3 were subjected to higher mutational pressure and could generate mutations as high as 57.1% to 71.4%. The M and N proteins, on the other hand, have a higher degree of conservation and are not mutated [116].

In general, the structural protein S is recognized as the most genetically diverse protein with the highest mutation frequency, and the S gene is often used to construct a phylogenetic tree to study the evolutionary relationship between virus strains. Firstly, the state of S protein N-linked glycosylation may affect viral replication and cellular tropism of viral pathogens. The S1 N-terminal region is the region with the largest number of amino acid variations [81,125]. Variations in this region may lead to the generation or elimination of potential N-glycosylation sites [126]. It is not difficult to see that the N-terminal region of the S protein may be an important area in which to study the genetic correlation between virus strains. The glycosylation site at residue 725 of the S protein is also a common site of frequent loss [124], and its function is still unknown. Secondly, variation in the N-linked glycosylation site of the S protein may affect the antigenicity of PEDV [111]. The S protein epitope SS2 (^748^YSNIGVCK^755^) and epitope 2C10 (^1368^GPRLQPY^1374^), which can induce neutralizing antibodies against PEDV, were conserved in all the Chinese field strains [127]. However, the epitopes COE (aa 499–638) and SS6 (^764^LQDGQVKI^771^) show diversity in most field strains and are mutation-prone areas in the epitope regions [127]. Variations on the epitopes COE and SS6 of the *S* gene may be important references for the development of attenuated vaccines.

ORF3 and S genes have the most insertions and deletions of nucleotide fragments, and may be the two most heterogeneous structures. Much of the focus has been on ORF3 insertions or deletions, and the effects of mutations in ORF3 on virulence or other properties have rarely been examined. As a controversial accessory protein for virulence [122,128], the amino acid variation caused by its nucleotide mutation may be more related to the adaptability of the growth environment.

The E protein, like the S, is a cellular marker of PEDV growth adaptation and virulence attenuation [126]. However, genetic and phylogenetic analyses of the E gene have been rarely performed. Except for the attenuated DR13, the E gene of all Chinese PEDV strains has very high conservation [9]. So far, no relevant studies have verified the function of the E protein in virus virulence.

The M protein is involved in the viral envelope, and interacts with the virion core, playing an important role in the viral assembly process [129]. The M gene sequence analyses of 31 PEDV isolates obtained in Thailand and of six PEDV isolates obtained in China both showed that there were no insertions and deletions in the entire M gene, and the glycosylation sequence Asn-Phe-Thr (NFT) in the M gene was also highly conserved [21,130]. The relationship between M gene heterogeneity and virus pathogenicity has not yet been well examined.

The N protein is an important component of phylogenetic analysis [131,132,133], involved in packaging the viral genome to facilitate viral assembly [134]. The N gene is also highly conserved; none of the early Chinese isolates and the 15 Chinese field samples had insertions or deletions in the N gene, and the N-terminal part of the protein was more conserved than the C-terminal part [132,135]. PEDV strains with mutations in the N gene may evade the established host immune responses [136], enhancing the growth adaptability of the virus itself.

### 2.3. Potential Key Amino Acids

As a key protein that determines the virulence of PEDV, when the attenuated strain does not produce representative variants of deletion or insertion, the effect of S gene mutations on virus virulence and growth adaptability is particularly important, and cannot be ignored. On the one hand, mutations in the S gene may impair the recognition of porcine receptors and reduce the replication effect of PEDV in vivo [137]. On the other hand, mutation of the S gene may be a positive selection, leading to differences in its immunogenicity, which in turn diversifies the neutralization profiles of the virus [138]. To find functional amino acid sites that may be closely related to virus pathogenicity and growth fitness, in Table 3, we summarize the amino acid mutations accumulated in five cell-adapted attenuated strains and one field attenuated strain.

In these PEDV strains, several potentially important amino acid mutations have been elucidated in a review [137]. Amino acid mutations with different properties accumulate in the virus strains with an increasing number of passages, while the mutational spectrum is stable over time, which may increase the stability of the virus in the host. The accumulation of amino acid substitutions is accompanied by drastic changes in amino acid properties, especially mutations in the S1 subunit of the S protein, more than 93.3% of which result in transitions between hydrophilic and hydrophobic amino acids and changes in the charged properties of amino acids. However, the number of these transitions between hydrophilic and hydrophobic amino acids is essentially symmetrical in each strain, which may maintain the biochemical stability of viral proteins. As previously mentioned, the S1 subunit of the S protein is an important factor in determining PEDV virulence [118]. The PC220A-P120, PT-P96, YN144, FJzz1-F200, and OH851 strains were all mutated at domain 0 of the S1 subunit, which has the effect of binding sialic acid receptors on host cells to attach to virions [142]. Mutations in this domain may disrupt binding to sialic glycans and thereby reduce PEDV entry. The domains S A, S B, and S CD of the S1 subunit of the S protein have many B-cell epitopes that induce neutralizing antibodies [142,143], and mutations in these domains may affect the recognition of neutralizing antibodies. Especially in the CT-P120 and PT-P96 strains, the F636R and F636S mutations on the epitope COE are likely to reduce the reactivity of virus-neutralizing (VN) antibodies. The S2 subunit, which contains major immunodominant neutralizing epitopes [144], has a higher mutation frequency than the S1 subunit. More than 63% of mutations in the S protein occur in the S2 subunit. However, the mutations in the S2 subunit are milder. In YN144, PC22A-P120, and CT-P120 strains, 50–60% of the amino acid mutations did not change their biochemical properties, and the rest were changed in respect to charge properties. In PT-P96 and FJZZ1-F200 strains, there are still 85% of amino acid mutations with drastic changes in amino acid properties (charged, hydrophilic or hydrophobic), which may tolerate higher mutation pressure. Frequent mutations on the S2 subunit may be the main reason for the failure of neutralizing antibodies. Taken together, the attenuated virulence of these strains may be due to mutations in the *S* gene that impair the ability of receptor recognition and binding.

Studies have reported that trypsin can promote cell fusion and viral infection by cleaving the S protein [145]. Efficient viral infection of strains that result in enhanced cellular adaptability through serial passaging is generally more dependent on the presence of trypsin, and mutations in the S2 subunit can control the trypsin dependence of PEDV [146]. In these cell-adapted strains, mutations in the S2 subunit may be closely related to cleavage by trypsin [147]. It has been confirmed that the S888R (R885) mutation may promote the cellular adaptation of the virus by adding a new trypsin cleavage site [146]. Since trypsin cleaves only at arginine and lysine residues, we can speculate that the ^888–889^SG →RR, F1015L, K1027R, V1242L, and I1305L mutations in these strains might increase trypsin cleavage, to promote the cellular adaptation of PEDV. In addition, common mutations in these attenuated strains may also be a marker of cellular adaptation [146]. From Table 3, we can see that the variant at residue 144 of domain 0, the D265A substitution in domain S A, and the S888R and C1355F substitutions in the S2 subunit, all have higher mutation frequencies, which may be closely related to cell adaptability.

Unlike those cell-adapted strains, amino acid variations on the field strain OH851 were more associated with virulence. Piglets infected with the new variant OH851 with insertions and deletions in the S gene exhibited few or no clinical symptoms [51]. The field variant strain GER/L00719/2014 found in Germany and Portugal, although 99.5% similar to the American OH851 strain [13], had typical severe watery diarrhea symptoms and high mortality [64,67]. Sequence alignment revealed that changes in clinical pathogenicity may be due to the variations in key amino acids that affect virulence. Compared with cell-adapted strains, the variation in the field strain OH851 deserves more attention, because it is a likely target for the development of attenuated vaccines.

## 3. Virus-Host Interaction

PEDV prefers to infect and replicate in the swine intestinal tract, including the villous epithelial cells of the small intestine and jejunum, and the surface epithelial cells of the caecum and colon, of which the jejunum and ileum are the main sites of infection [43,148,149,150]. Infection with PEDV promotes fusion between cells, leading to continuous necrosis of cells, resulting in severe diarrhea and even death in piglets [7]. At the same time, PEDV is sensitive to the porcine liver, resulting in abnormal cholesterol metabolism genes, and disrupting the metabolic homeostasis of cholesterol [151,152].

A comprehensive understanding of the infection mechanism by which PEDV invades host cells and achieves replication has important implications for the prevention and control of viral infection. Firstly, PEDV relies on serine proteolysis and low pH to enter cells through endocytosis, which is the entry mechanism by which PEDV infects the host [153]. Secondly, PEDV mainly utilizes viral components to evade the host’s innate immune responses during host infection. It is well known that after the virus invades the host cells, interferon (*IFN*) plays an important role as a key signaling molecule to block virus infection [154], especially type I *IFNs*, which clear the virus by activating various immune cells and costimulatory molecules [155].

To better understand the biopathogenesis of PED, here, we map the network of PEDV-host interactions (Figure 3), and further elucidate the roles of PEDV components during viral infection. Moreover, host factors, especially functional receptors, may be potential antiviral targets for the effective control and elimination of PEDV. Here, we extend the map of pro/anti-viral host factors of PEDV and summarize the roles of the host factors during PEDV infection (Figure 4).

### 3.1. PEDV Proteins That Interact with Host Factors

Cell entry of the coronaviruses mainly depends on the interaction of the S protein with the receptor. The cleavage and activation of the S protein may depend on the hydrolysis of S protein by serine proteases at the S1/S2 junction and adjacent to the fusion peptide in the S2 region, during the virus entry process [153,156]. The S protein enhances TfR1 internalization by interacting with the extracellular domain of TfR1, facilitating viral entry [157]. The S1 subunit of the structural protein S is a key region for the recognition and binding of cellular receptors [158]. Both the C-terminal domain (S1-CTD) and N-terminal domain (S1-NTD) of the S1 subunit of coronaviruses can interact with receptors as RBDs [159]. In a variety of coronaviruses, S1-CTD recognizes and binds to the angiotensin converting enzyme 2 (ACE2), dipeptidyl peptidase 4 (DPP4), and aminopeptidase N (APN) to mediate the entry of SARS-CoV, human coronavirus NL63 (HCoV-NL63), Middle East respiratory syndrome coronavirus (MERS-CoV), transmissible gastroenteritis virus (TGEV) and porcine respiratory coronavirus (PRCV) [159,160,161,162,163,164,165,166,167,168]. Meanwhile, S1-NTD, as RBDs, interacts more with the carcinoembryonic antigen cell adhesion molecule 1 (CEACAM1) or sugar receptors to promote the attachment and entry of viruses such as mouse hepatitis virus (MHV) and bovine coronavirus (BCoV) [169,170,171]. Similar to other coronaviruses, the RBDs of the receptor pAPN are located within the S1-CTD of PEDV, while the S1-NTD of PEDV is considered to be the RBDs of sialic acid glycans [72,172]. One study showed that the S protein inhibits *IFN*-mediated innate immune responses by binding to the epidermal growth factor receptor (EGFR) to increase PEDV infection [173]. As a transmembrane glycoprotein, EGFR may bind to S1-CTD, to promote virus entry, but the RBDs of the S protein that interacts with it remain to be verified.

The E and M proteins inhibit type I *IFN* immune responses by directly interacting with IRF3 and *IFN* regulatory factor 7 (IRF7), respectively [89,174]. In addition to evading innate immune responses, the E protein induces endoplasmic reticulum (ER) stress, upregulates IL-8 expression, and may play a role in inhibiting host-cell apoptosis [87]. It has been confirmed that the M protein interacts with five proteins (RIG-I, PPID, NHE-RF1, S100A11, CLDN4), among which the M protein may inhibit the proliferation of PEDV by interacting with S100A11 or PPID [175].

The structural protein N plays an important role in promoting viral replication and anti-innate immune responses. In TGEV, the N protein is not essential for RNA replication but is a key protein required for transcription [100]. While the role of PEDV N protein in RNA synthesis has not been elucidated, there has been a lot of indirect evidence for the role of N in PEDV replication. One piece of evidence is that the N protein colocalizes and interacts with nucleophospholipid (NPM1) in the nucleolus, promoting viral replication and inhibiting the apoptotic capacity of cells [104]. Similarly, the interaction of the N protein with p53 promotes viral replication mainly by inducing arrest in the S phase of the cell cycle [176]. Another piece of evidence is that the N protein interacts with heterogeneous nuclear ribonucleoprotein A1 (hnRNP A1), an RNA-binding protein involved in pre-mRNA splicing and mRNA nucleo-cytoplasmic export in the nucleus [177], promoting PEDV replication [178]. In addition, the PEDV N protein has many mechanisms to evade the host immune responses, one of which is the most classical anti-*IFN* immune response: the N protein antagonizes the production of *IFN*-λ3 by blocking the nuclear translocation of NF-κB [179]; the N protein targets TANK-binding kinase 1 (TBK1) through direct interaction, which competitively inhibits the binding of TBK1 to IRF3, resulting in the inhibition of IRF3 activation and the subsequent production of type I *IFN* [102]. Another novel mechanism for evading host immune responses is the inhibition of HDAC1, an important regulator of innate immunity: the N protein in the nucleus interacts directly with Sp1, an important transcriptional regulator of histone deacetylase (HDAC) expression, indirectly inhibits the replication and transcription of multiple innate immune effectors regulated by HDAC1 [180,181].

ORF3 is mainly localized in the ER and triggers ER stress responses related to apoptosis or autophagy [182]. Accumulation of ORF3 is also found in the Golgi apparatus (Golgi), which is transported from the ER to the Golgi region via the exocytosis pathway [120]. We also found the presence of ORF3 on the surface of infected cells, and interacting with the S protein at the plasma membrane [183], which may be related to the ability to regulate viral replication, where the ^170^YLAI^173^ motif in the C-terminal part of ORF3 protein is crucial for the transport of ORF3 from the ER to the plasma membrane [120]. ORF3 inhibits *NF*-κB activation by inhibiting the phosphorylation and expression of nuclear factor P65 and interfering with the nuclear translocation of P65, thereby reducing the production of proinflammatory cytokines IL-6 and IL-8 [184]. Meanwhile, ORF3 interacts with IκB kinase β (IKBKB), induces IKBKB-mediated *NF*-κB promoter activity, and activates type I *IFN* induction, but inhibits Poly I: C mediated type I *IFN* production and induction [185]. These findings highlight the complex role of ORF3 in immune signaling and virus–host interactions.

The nonstructural proteins nsps encoded by ORF1a/b have distinct anti-host immune responses, possibly regulating viral replication by enhancing the translation of downstream ORFs [186]. Among them, nsp1 is the most potent suppressor of proinflammatory cytokines, interfering with *IFN*-induced immune processes [187,188]. Unlike the E protein interfering with IRF3 nuclear translocation, nsp1 does not interfere with IRF3 phosphorylation and nuclear translocation, but blocks the assembly of IRF3 and CREB-binding protein (CBP) enhanceosome by degrading CBP to suppress the innate immune responses of type I *IFN* [97]. Besides this, nsp1 also interferes with the nuclear translocation of interferon regulatory factor 1 (IRF1), and suppresses IRF1-mediated type III *IFN* immune responses [187]. The nsp2 acts through a novel mechanism of non-*IFN*-mediated host immune responses, which interacts with the innate antiviral factor FBXW7 and hinders the activation of the host innate antiviral response by targeting the ubiquitin-proteasome-mediated degradation of FBXW7 [189]. Interestingly, nsp4 is involved in the inflammatory response of the host, directing the expression of proinflammatory cytokines and chemokines, possibly inhibiting PEDV replication in vitro [190]. PEDV nsp5, encoding 3C-like protease, is another *IFN* antagonist as well as the N protein. Consistent with the foot-and-mouth disease virus (FMDV), hepatitis A virus (HAV), and PRRSV [191,192,193], PEDV nsp5 cleaves the NF-kappaB essential modulator (NEMO), disrupting type I *IFN* signaling [194]. The nsp6 induces autophagy through the PI3K/Akt/mTOR signaling pathway and has a similar function to its downstream nsp9 in promoting PEDV replication [195,196]. During PEDV infection, PEDV degrades the signal transducer and activator of transcription 1 (STAT1) proteins in a protease-dependent manner, to interfere with the type I *IFN* signaling pathway [197]. However, the functional protein of PEDV that degrades STAT1 remains unclear. There is only limited evidence that nsp7 is related to STAT1, and nsp7 can interact with the DNA-binding domains of STAT1/STAT2, blocking the nuclear translocation of STAT1, and further antagonizing *IFN*-α-induced JAK-STAT signaling [198]. In addition, nsp12, an RNA-dependent RNA polymerase, and nsp13, a helicase, participate in the replication process of PEDV by promoting the release of viral RNA [196,199]. The G-N-7 methyltransferase (G-N-7 MTase) activity of PEDV nsp14 plays an important role in regulating PEDV replication and type I and type III *IFN* immune responses [200]. Unlike the N protein sequestered binding of TBK1 and IRF3, nsp15 utilizes endoribonuclease (EndoU) activity to degrade the RNA of TBK1 and IRF3, to inhibit their mediated type I *IFN* response. The three residues (H226, H241, and K282) of nsp15 that affect the activity of EndoU may be the key amino acids for the inhibition of the antiviral activity [201]. The methyltransferase nsp16 is a more potent regulator of immune-related genes, which relies on KDKE tetrad to regulate the 2′-O-MTase activity, not only reducing *IFN*-β production but also inhibiting IRF3 phosphorylation, effectively modulating the host antiviral response to multiple viruses including PRRSV, vesicular stomatitis virus (VSV), and PEDV [202].

Collectively, each protein component of PEDV plays an important role in enhancing viral replication and assembly and evading the host’s innate immune responses. In the case of some nonstructural proteins, such as nsp1, nsp5, nsp6, and nsp13–16, although their roles in evading innate immune signaling have been elucidated, whether they interact with host chaperones remains to be investigated. The analysis of interacting proteins may be important for the elucidation of the molecular mechanisms that promote viral proliferation and evade host immune responses.

### 3.2. Proviral Host Factor

Host factors required for coronavirus replication are the main targets of antiviral drugs, especially pan-coronavirus host factors. Unraveling the host factors required for coronavirus to infect cells is critical to the development of antiviral drugs.

Each coronavirus has individual specific functional receptors. However, due to limitations such as frequent variants of PEDV and the lack of susceptible host cell lines, especially porcine-derived ones, the research on the infection mechanism of the PEDV virus and the mining of functional host factors have become difficult. difficult. In the process of infecting the host, the virus usually utilizes the components of the host cell to promote the entry, replication, and assembly of the virus, which we generally refer to as proviral host factors. Proviral-host factor inhibitors have also been a commonly used means of antiviral therapy. In Table 4, we summarize the proviral host factors of PEDV and their important functions during virus infection, and describe their mechanism of proviral effect in host cells.

APN (CD13) is an ectoenzyme with membrane binding and signaling functions [204] and is a functional receptor for various coronaviruses such as human coronavirus 229E (HCoV-229E) and TGEV [168,235]. Porcine APN (pAPN) is a functional receptor of PEDV [203], and PEDV S1-NTD-CTD can effectively bind to domain VII of pAPN for entry into the host cells [72,236]. However, recent studies have found that APN is not essential for PEDV entry into cells [237], and the claim that pAPN acts as a functional receptor for PEDV is inaccurate. The increased susceptibility of pAPN transgenic mice to PEDV may be explained by pAPN promoting PEDV infection through its protease activity [238,239].

Sialic acid (SA), mainly the terminal component of glycoproteins and glycolipids, a monosaccharide derivative, exists on the cell surface and is a cofactor for viral attachment and entry into cells [240]. There are three forms of SA commonly found in mammals: 5-N-acetylneuraminic acid (Neu5Ac), 5-N-glycolylneuraminic acid (Neu5Gc) and 2-keto-deoxynonulosonic acid (Kdn) [241]. Sialylglycans are the functional receptors for influenza viruses such as avian and human infectious bronchitis coronavirus (IBV) and influenza A virus (IAV), but avian and human influenza viruses generally have different SA binding preferences [242,243]. SA facilitates the infection of multiple coronaviruses, among which MERS-CoV and TGEV preferentially bind Neu5Ac [244,245]. PEDV as betacoronaviruses (β-CoVs) also utilizes Neu5Ac as a sugar receptor. However, the binding ability of S protein to SA varies with PEDV strains [73].

Tight junction (TJ) proteins are multiprotein complexes formed by several different integral membrane proteins [246]. A variety of viruses use components of the TJ proteins as receptors to enter host cells through internalization pathways, including the coxsackie and adenovirus receptors (CAR) and the human occludin receptor [247,248,249]. The TJ protein occludin plays an important role in maintaining and repairing the intestinal mucosal barrier [208]. Recently, studies have shown that occludin is not effective at the attachment of PEDV, but occludin acts as a cofactor for PEDV entry into host cells, whose internalization is closely related to virus entry [250].

Nucleolar phosphoprotein nucleophosmin 1 (NPM1), which predominantly resides in the nucleolus, is a multifunctional protein that plays an important role in RNA transcription, ribosome assembly, DNA replication and repair, nuclear export, and cell growth regulation [209]. NPM1 promotes infection by a variety of viruses, including the human immunodeficiency virus type 1 (HIV-1) [251], Japanese encephalitis virus (JEV) [252], adenovirus [253], herpes simplex virus 1 (HSV-1) [254], Epstein–Barr Virus (EBV) [255], and Schmallenberg virus (SBV) [256], function at all stages of viral infection. As a proviral factor, NPM1 binds to the cap proteins of porcine circovirus types 2 and 3 (PCV2 and PCV3) to promote nucleolar localization and viral replication [257,258,259]. However, there is also evidence for NPM1 as an antiviral factor that limits the replication of the chikungunya virus (CHIKV) [260]. NPM1 binds to the N protein of PEDV and colocalizes in the nucleolus, ultimately promoting viral replication mainly by enhancing cell growth and survival [104]. The binding of NPM1 to other proteins of PEDV and its role in viral infection remains to be further explored.

Heat shock protein 70 (HSP70) is involved in the posttranslational folding of the protein, membrane protein localization, assembly and disassembly of the protein complex [261,262], and promotes viral replication during RNA virus infection [263]. HSP70 plays a proviral role by binding to the M protein during PEDV infection [210]. However, Hsp70 exerts antiviral effects by inhibiting the nuclear export of the influenza virus ribonucleoprotein complex (vRNP) and the replication process of VSV infection [264,265]. HSP70 has positive and negative dual regulatory roles on different virus types, showing the complexity of virus-chaperone protein interactions.

The epidermal growth factor receptor (EGFR) is a tyrosine kinase receptor that acts as a cofactor for viral entry into the host cells during hepatitis C virus (HCV), TGEV, and IAV infection [266,267,268,269]. EGFR signaling is a central pathway required for SARS-CoV-2 replication [270], and EGFR inhibition has emerged as a novel strategy to suppress endogenous antiviral defenses during host infection by respiratory viruses and PEDV [173,270]. However, at what stage EGFR plays a role in PEDV infection of host cells, has not been elucidated.

### 3.3. Antiviral Host Factor

In addition to host factors with proviral effects, some host factors resist the invasion of viruses by conferring protection on cells against virus infection, which are generally called antiviral host factors. Antiviral factors are positive regulators of the host’s innate immune system, and become potential targets for antiviral therapy. In Table 4, we summarize the antiviral host factors of PEDV and describe their antiviral effects in host cells.

Bone marrow stromal cell antigen 2 (BST2), also known as CD317, is an *IFN*-induced type II transmembrane glycoprotein [205]. BST2 inhibits the release of a large number of enveloped viruses, such as human HCoV-229E [271], human HCV [272], Dengue virus (DENV) [273], JEV [274], HIV [275,276], Ebola virus [277], Marburg virus (MARV) and Lassa virus (LASV) [278], HSV-1 [279], human cytomegalovirus (HCMV) [280] and other types of viruses. Studies have shown that BST2 combines with PEDV N protein to achieve protein degradation, thereby inhibiting the biological function of the N protein, while the role of BST2 in viral envelope release has not been elucidated.

Vacuolar Protein Sorting Associated Proteins 36 (VPS36) together with Vps22 and Vps25 form the ESCRT (Endosomal Sorting Complexes Required for Transport)-II complex [281], which plays a role in the multivesicular body (MVB) pathway of protein sorting and degradation, leading to the degradation of proteins in lysosomes [206]. VPS16 and VPS35 of the VPS family promote the infection of human coronavirus OC43 (HCoV-OC43), HCoV-229E, and SARS-CoV-2 coronaviruses [282,283]. However, VPS36 was found to interact with ORF3 during PEDV infection, inhibiting PEDV replication by promoting ORF3 degradation [213]. The VPS family may play different roles in the viral infection process.

Cholesterol-25-hydroxylase (CH25H) is a hydroxysterol belonging to the interferon-stimulated gene (ISG) family [284]. CH25H and its enzymatically catalyzed cholesterol product, 25-hydroxycholesterol (25HC), have diverse biological functions, including the regulation of cholesterol metabolism, the dual role of suppressing or increasing inflammation, and the regulation of B- and T-cell-mediated immune responses [214]. CH25H and 25HC also have broad antiviral activity, inhibiting the infection of VSV [285], rabies virus (RABV) [286], LASV [287], ZIKV [288], human rhinovirus (HRV) [289], porcine pseudorabies virus (PRV) [290], and PRRSV [291,292] through multiple mechanisms. Recent evidence suggests that CH25H and 25HC play an important role in hindering the entry of PEDV virions [215]. As an antiviral factor, CH25H may be an important target for broad-spectrum antiviral therapy.

Ras-GTPase-activating protein-binding protein 1 (G3BP1) is a multifunctional RNA-binding protein involved in various biological functions such as RNA recognition, mRNA turnover and translation, cell proliferation, apoptosis, and differentiation [293]. G3BP1 is also a key component in the assembly of SGs [294]. In response to stimuli, host cells form stress granules (SGs) to prevent viral protein synthesis and regulate viral replication [295]. SGs are more antiviral, and a variety of viruses including many RNA viruses [296,297,298,299,300,301], enteroviruses [302,303], coronaviruses [304], influenza viruses [305], and reoviruses [306,307], possibly inhibit the formation of SGs by cleaving G3BP1 or disrupting the eIF4GI-G3BP1 interaction. Similarly, G3BP1 has an antiviral role in PEDV infection, and PEDV promotes viral replication by inducing the degradation of G3BP1 [216,217]. however, CHIKV of the alphavirus genus has evolved a different mechanism to promote efficient replication by utilizing the SGs component G3BP1 [308]. Overall, G3BP1 functions as an antiviral factor in most viral infections.

The ubiquitin ligase FBXW7 is generally regarded as a tumor suppressor capable of targeting various oncogenic proteins for negative regulation [309]. Endogenous FBXW7 significantly inhibited human HCV replication [310], and a recent report demonstrated that FBXW7 is an innate antiviral factor. The nonstructural protein nsp2 mediates the degradation of FBXW7 by interacting with FBXW7, ultimately suppressing the host antiviral response [189]. Inhibiting the degradation of FBXW7 protein may be an effective strategy for antiviral therapy.

A disintegrin and metalloprotease 17 (ADAM17/CD156b), a member of the metalloprotease superfamily, is responsible for the cleavage of various cell surface proteins and may play a role after viral initial binding and release [219,311]. ADAM17 may have dual roles during viral infection. On the one hand, ADAM17 induces the replication of human papillomavirus (HPV) [311] and HIV-1 [312]. ADAM17 binds directly to E2 protein and plays a key role in CSFV entry [313], however regulation of ADAM17 activity does not affect SARS-CoV entry [314]. On the other hand, ADAM17 downregulates the expression of CD163, a functional receptor for PRRSV, and blocks viral entry [315]. ADAM17 also inhibits PEDV infection by regulating APN expression [316]. The mechanism of action of ADAM17 in viral infection requires more research evidence.

The L subunit of human eukaryotic translation initiation factor 3 (eIF3L) is one of the subunits of eukaryotic initiation factor (eIF3), which interacts with other eIF3 subunits and may play a role in increasing the physical stability of eIF3 [220]. The other subunit of eIF3, eIF3f, can specifically interfere with the 3′ end processing of mRNA, thereby inhibiting HIV-1 replication [317]. The eIF3L can interact with the M protein of PEDV and significantly inhibit virus replication, but how eIF3L inhibits virus infection remains to be further explored [221]. The eIF3L interacts with the NS5 protein of yellow fever virus (YFV) but has a weak role in inhibiting viral replication. During viral infection [318], more attention may be paid to the role of eIF3L in viral translation.

Poly(A) Binding Protein Cytoplasmic 4 (PABPC4) is a nuclear-cytoplasmic shuttle protein that interacts with the N protein of PEDV, SARS-CoV-2, MHV, IBV, and PDCoV multiple coronaviruses [222]. PABPC4 may inhibit coronavirus infection by degrading the N protein through the cargo receptor NDP52-mediated selective autophagy [319,320]. PABPC4 may be a broad anti-coronavirus host factor, which has important implications for the selection of antiviral targets.

CD44 is a transmembrane glycoprotein that plays an important role in cell adhesion, growth factor regulation, and signaling such as RTKs, Met, members of the TGFβ family, CXCL12 and its induced CXCR4 [321,322]. Furthermore, CD44 promotes tumorigenesis as a major cell surface receptor for hyaluronic acid (HA) [323]. The presence or absence of CD44 does not affect poliovirus replication [324], and its role in viral infection has also been overlooked. However, recent studies have shown that CD44 is an antiviral host factor that inhibits PEDV infection by activating *NF*-κB signaling [224]. Similarly, the Goujon C group identified CD44 as a novel anti-SARS-CoV-2 protein factor with potent activity, which has a certain impact on the infection of other coronaviruses such as HCoV-229E and MERS-CoV [325]. CD44 has broad anti-coronavirus activity, which may restrict viral entry by affecting viral internalization [325]. Similarly, chicken CD44 acts as a cellular receptor for the infectious bursal disease virus (IBDV), promoting IBDV binding and entry in B lymphocytes [326]. However, CD44 has less effect on influenza A virus IAV infection [325]. The complex role of CD44 in viral infection of different families and genera is highlighted.

Interleukin-11 (IL-11) is an inflammatory cytokine that plays an important role in inflammation suppression, anti-apoptosis, epithelial regeneration, and fertility [225,226,227,228]. There is evidence that IL-22 inhibits infection by a variety of viruses, including porcine enteric coronaviruses (PEDV, TGEV), porcine rotavirus (PoRV), human respiratory syncytial virus (RSV), and HIV-1 [229,327,328,329]. Both IL-11 and IL-22 promote the inhibition of PEDV by activating the STAT3 signaling pathway, and are two important antiviral factors during viral infection.

Mucin 2 (MUC2) is a protein secreted by goblet cells that constitutes the intestinal mucus layer that protects the intestinal epithelial cells [330,331]. MUC2 mainly has the function of maintaining the intestinal biological barrier. The expression of MUC2 mRNA was induced in PRRSV infection, which played a role in the maintenance of intestinal integrity [332]. Similarly, in PEDV infection experiments, the role of MUC2 in the antiviral was determined [234].

In addition to host genes involved in the viral infection process, MicroRNA-221-5p, a non-coding RNA involved in post-transcriptional regulation of genes, can also target the 3′UTR of the viral genome and activate the *NF*-κB signaling pathway to inhibit the replication of PEDV [333,334]. Noncoding RNA processing may become a new idea for antiviral therapy.

## 4. Conclusions and Perspectives

The high morbidity and high mortality that PEDV brings to piglets results in a rapid transmission of PEDV in pig populations, which has brought huge economic losses to the global swine industry. Considering the hazards caused by PEDV, preventive immunization is administered by vaccinating pregnant sows. However, due to the high heterogeneity of PEDV, there are currently no effective and safe vaccines to deal with the threat posed by PEDV. Exploring the genetic variation of PEDV and its virulence changes will help us to have a deeper understanding of PEDV and take more efficient diagnosis and treatment measures, accordingly. Similarly, given the problem that there are no efficient vaccines for PEDV treatment, exploring the pathogenesis of PEDV and mining the host factors that interact with PEDV will help us to find potential therapeutic targets, and then develop new prevention and control strategies. Mining host factors as the candidate targets for antiviral therapy, potentially addresses the challenges posed by PEDV genetic diversity to vaccine development. Based on the current state of research, future research should focus on finding the crucial functional receptor for PEDV or identifying key therapeutic targets, and finally developing alternative strategies to target the host proteins or regulators of immune responses, to control this pandemic.

## Figures and Tables

**Figure 1 viruses-14-02434-f001:**
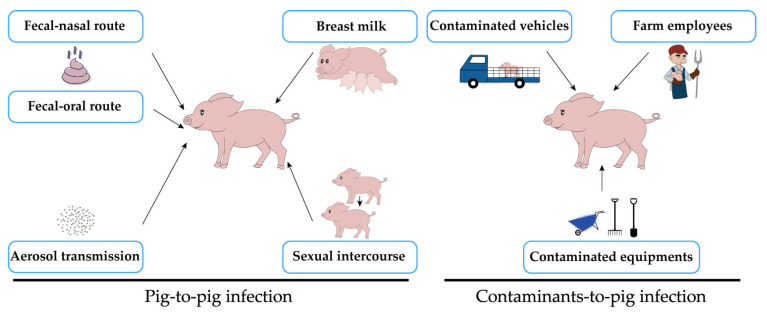
Several transmission routes of PEDV infection. Black arrows indicate transmission routes of PEDV.

**Figure 2 viruses-14-02434-f002:**
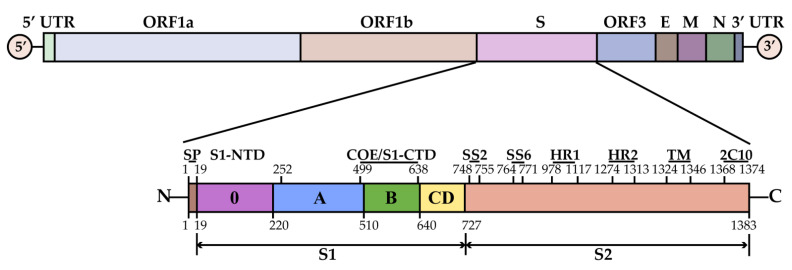
Schematic diagram of the PEDV genome structure. Non-structural proteins, including nsp1–nsp16; structural proteins, including spike (S), envelope (E), membrane (M), and nucleocapsid (N) proteins; accessory proteins, ORF3. The spike protein consists of two subdomains, S1 and S2. The S1 subunit can be further subdivided into 6 domains, including signal peptide (SP, 1–18 aa, brown), domains 0 (19–219 aa, purple), domains A (220–509 aa, blue), domains B (510–639 aa, green), domains C and D (640–728 aa, yellow). The S1 subunit contains two receptor binding domains (RBDs): the N-terminal domain (S1-NTD, 19–252 aa) and the C-terminal domain (S1-CTD, 509–638 aa). The S2 subunit contains 3 domains, including two heptapeptide repeat domains (HR1 and HR2, 978–1117 aa and 1274–1313 aa) and a transmembrane domain (TM, 1324–1346 aa).

**Figure 3 viruses-14-02434-f003:**
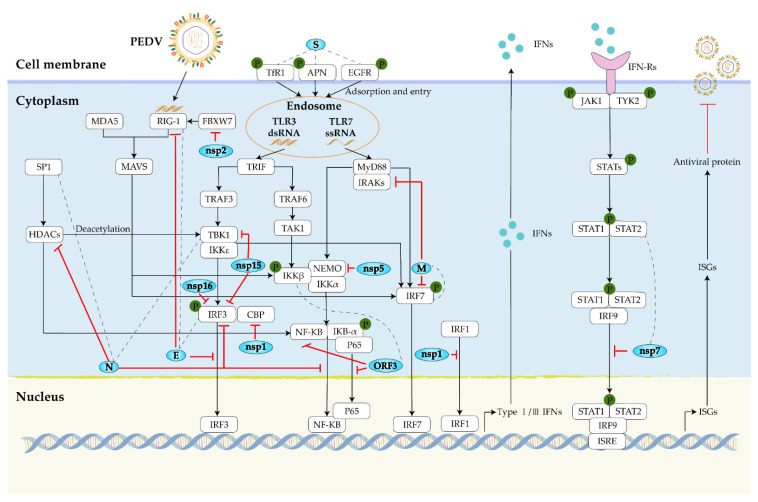
Schematic diagram of PEDV evading the host innate-immune signaling pathway. On the one hand, PEDV infection activates the host innate-immune signaling pathway. On the other hand, PEDV proteins exert inhibitory effects at different stages of the host innate immune response. Interaction of PEDV components with host proteins is indicated by dotted lines; downstream activation is indicated by black arrows; inhibition or degradation of target genes is indicated by red lines ending in dashes.

**Figure 4 viruses-14-02434-f004:**
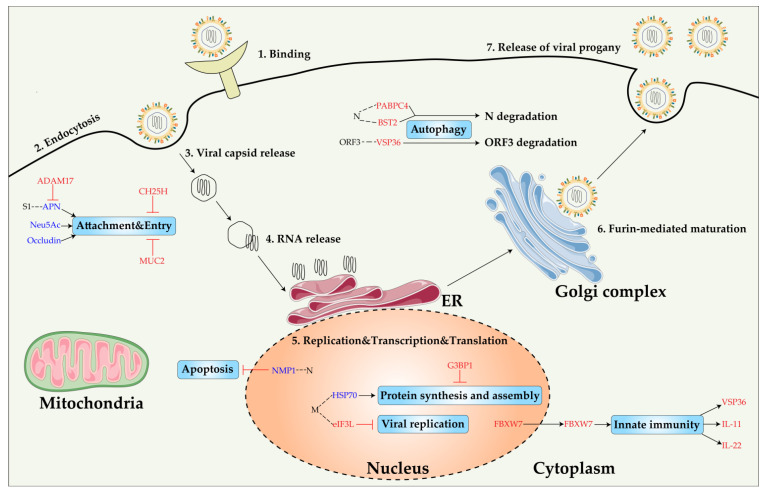
Schematic illustration of PEDV infection in host cells and the role of pro/antiviral host factors in PEDV replication. There are four main stages in the PEDV life cycle: attachment, entry, replication, and release. Interaction of host proteins with PEDV proteins is indicated by dotted lines; proviral host factors are indicated in blue text; antiviral host factors are indicated in red text; downstream activation is indicated by black arrows; inhibition or degradation of target genes is indicated by red lines ending in dashes.

**Table 1 viruses-14-02434-t001:** Representative strains and distributions of each PEDV genotype.

Genotype	Variation	Representative Strains	Distribution	Reference
GIa	Classical strains	CV777; DR13; LZC; CH-S; AVCT12; SM98; CHM2013	Asia–China, Korea, Thailand; Europe–Belgium, Russia	[19,20,21,22]
GIb	S-Indel vaccine strains relative to GIa	JS-2004-2; attenuated CV777; attenuated DR13; SD-M; SC1402; ZJ08; SQ2014; AH-M	Asia–China, Korea, Japan, Thailand	[1,19,23,24,25]
GIIa	Recombined non-S INDEL variant strains	AH2012; LZW; ZMDZY; GDZQ; FJZZ-9; GD-B; KNU-1305; Tottori2; MYG-1; Colorado; PC21A; PC22A; PC177; TC-PC177; MN; Kansas-166; Minnesota79; MEX/104/2013	Asia–China, Korea, Japan; North America–USA, Mexico	[6,19,22,23,26,27,28]
GIIb	Recombined non-S INDEL variant strains	AJ1102; LC; AH2012-12; YN1; CHGD-01; GD-1; GDS01; GD-A; ZJCZ4; HNPJ	Asia–China	[6,19,26,27,29]
GIIc	S-INDEL strains recombined between GIa and GIIa	ZL29; CH/HNQX-3/14; KNU-1406-1; OH851; Iowa106; Minnesota211; IL20697; GER/L00862/2014; 15V010; FR/001/2014	Asia–China, Korea; North America–USA; Europe–Germany, Romania, Austria, Belgium, Italy, France, Hungary, Slovenija	[1,6,19,22,25,29,30,31]

**Table 2 viruses-14-02434-t002:** Representative PEDV strains with protein variations that influence changes in viral virulence.

Genotype	Strain	ORFs	Major Variation	Pathogenicity	Reference
GIIb	icPC22A-KDKE^4A^-SYA	Nsp16, S	Nsp16 (quadruple alanine substitutions[K45A-D129A-K169A-E202A]), S (Y1378A)	Attenuated	[109]
GII	FJzz1-F200 (FJzz1)	S1-NTD	^55^I^56^G^57^E → ^55^K^56^Δ^57^Δ, ^877–878^SG →^877–878^RR	Attenuated	[110,111]
GII	Iowa106 (PC21A)	S1	Deletion at nt 176–186 and nt 416–418, 6-nt insertion at nt 474–475	Attenuated	[63,112]
GII	5-17-V (KF452323)	S1-NTD	Deletion at aa 23–229	Attenuated	[113]
GIIa	TC-PC177, icPC22A-S1Δ197 (PC21A, icPC22A)	S1-NTD	Deletion at aa 34–230	Attenuated	[44,74]
GIIb	icPC22A-icΔ10aa (icPC22A)	S-CTs	ΔYxxΦEKVHVQ	Attenuated	[114]
GIIb	FL2013 (AJ1102)	S-CTs	Deletion at aa 1385–1391	Attenuated	[1,115]
GIIb	KNU-141112-S DEL2/ORF3 (KNU-141112)	S, ORF3	S N-DEL2 (a combination of S C-DEL5/ORF3 N-DEL70, and ORF3 C-DEL88)	Attenuated	[116]
GIa	CHM2013, SM98, AVCT12 (CV777)	S-CTs, ORF3, M	S (deletion at nt 4144–4165), ΔORF3 (deletion at nt 1–30), M (4–aa [MLVL]) insertion at nt 36–37)	Mild	[55,117,118]
GIIb	17GXCZ-1ORF3d (17GXCZ-1ORF3c)	ORF3	Deletion at nt 172–554	Virulent	[119]
GIIa	HN2021 (HNCADC-2017)	ORF3	Deletion at nt 207–373	Virulent	[117]
GIb	attenuated DR13 (DR13)	ORF3, E	ORF3 (Deletion at nt 245–293), E (Deletion at nt 67–87)	Attenuated	[20,120,121]
GIIb	SH (LZW)	N	Deletion at aa 399–410	Virulent	[30]

S1 N-terminal domain: S1-NTD; S1 core neutralizing epitope: S1-COE; S C-terminals: S-CTs; ^55^I^56^G^57^E → ^55^K^56^Δ^57^Δ, deletion aa GE at residues 56 to 57.

**Table 3 viruses-14-02434-t003:** Amino acid changes on the S protein of PEDV attenuated strains.

ORFs		PC22A-P120 [139]	CT-P120 [140]	PT-P96 [141]	YN144 [47]	FJzz1-F200 [110]	OH851 [51,64]
S1	domains 0	^55–57^IGE→K–, I166V		T144I	∆144–145TG	^55–57^IGE→K–, F128Y	S88A, N130D, N132K
	domains A	Q454K, D466G, ^477H	D265A		D405G, G428A	D265A, D378N, T491R	T361A
	domains B/COE		F636R	F555S			
	domains CD				S723R		A694D
S2		V881F, Q893K, A971V, G1009V, F1015L, E1379 stop	S888R, C1363G	S888R, S969A, I1022S, K1027R, L1253R, C1355F, C1359F	T780N, Q826H, I1011V, I1305L, C1355F	K774N, ^888–889^SG →RR, L901V, N1010D, I1340T, C1355F	G1162S, V1242L

1. The amino acid changes of the attenuated strains CT-P120, PT-P96, YN144, FJzz1-F200, PC22A-P120, and OH851 are based on the nucleotide sequences of CT-P10, PT-P5, YN15, PC22A-P3, FJzz1-F20, and GER/L00719/2014, respectively. 2. The residue positions on each domain were referenced to the S protein of the PC22A strain (GenBank: KM392224.1). 3. S1 subunit domains 0, residues 19 to 219; domains A, residues 220 to 509; domains B/COE, residues 510 to 639; domains CD, residues 640 to 728; S2 subunit, residues 729 to 1387 [142]. 4. ^477H, aa H was inserted at residue 477.

**Table 4 viruses-14-02434-t004:** An overview of host factors.

	Host Factors	Function	Role in VIRAL Infection	Reference
Proviral factors	pAPN	Enzymatic cleavage of peptides; endocytosis; signal transduction	PEDV binds pAPN domain VII for entry into cells	[203,204]
	Neu5Ac	As a cell surface glycoprotein; immune regulation and recognition; viral interactions	As a sugar coreceptor for PEDV entry cells	[73,205,206]
	Occludin	A tight junction protein; signal transduction; innate immune regulation	As a PEDV entry cofactor	[207,208]
	NPM1	Ribosome assembly and chromatin remodeling; nuclear export; cell growth regulation	Interacts with N protein to promote PEDV growth	[104,209]
	HSP70	Protein folding; participates in cellular processes; promotes viral replication.	Interacts with M protein to regulate PEDV replication, viral protein synthesis, and assembly	[210]
	EGFR	Regulates endocytic transport; regulates sorting after internalization and endocytosis	Inhibits I-*IFN* response through STAT3-mediated signaling	[211]
Antiviral factors	BST2	Regulates the transport of secreted cytokines; *IFN*-inducing markers	Binds and degrades the N protein of PEDV to inhibit PEDV replication	[212,205]
	VPS36	Regulation of protein sorting and MVB biogenesis	Promotes degradation of ORF3 by interacting with ORF3	[206,213]
	CH25H	Regulates lipid metabolism, cholesterol homeostasis, inflammation, and immune responses	Inhibits entry of PEDV virions	[214,215]
	G3BP1	Involves RNA recognition, host mRNA turnover and translation, SG formation	Induces antiviral SG formation and impairs PEDV replication	[216,217]
	FBXW7	Regulates immune cells; tumor suppressor	Promotes host *IFN*-mediated antiviral response	[218]
	ADAM17	Mediates cleavage and cleavage of cell surface proteins	Inhibits PEDV infection by regulating APN expression	[219]
	eIF3L	Regulates the physical stability of eIF3 assembly	Inhibits PEDV replication by interacting with M protein	[220,221]
	PABPC4	An RNA processing protein that enhances translation and mRNA stability	Promotes degradation of N protein by interacting with N protein	[222]
	CD44	Regulates signal transduction and tumor growth; cell adhesion	Activates NF-κB nuclear translocation and enhances protective cytokine release	[223,224]
	IL-11	Regulates inflammation, apoptosis, epithelial regeneration and fertility	Anti-PEDV infection by activating the STAT3 signaling pathway	[225,226,227,228,229]
	IL-22	Inhibits pro-inflammatory responses; protects the host gut barrier; maintains tissue integrity	Anti-PEDV infection by activating the STAT3 signaling pathway	[230,231,232]
	MUC2	Regulates intestinal homeostasis	Regulates the replication of PEDV	[233,234]

## Data Availability

Not applicable.

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
