# Peer review of "Porcine Epidemic Diarrhea Virus: An Updated Overview of Virus Epidemiology, Virulence Variation Patterns and Virus–Host Interactions"

_viruses, 2022, doi:10.3390/v14112434_

Round 1

Reviewer 1 Report

The PEDV, a member of the Coronaviridae family, causes acute diarrhoea and dehydration in pigs. In this manuscript, the authors summary the genotyping, distribution, origin, virion structure and function of PEDV. They also described the effects of PEDV viral proteins on virulence change and growth adaptation of strains from the perspective of gene heterogeneity. The key receptors of PEDV are still unknown. Therefore, the uncovering the infection mechanism by which PEDV invades host cells and achieves replication is important for the prevention and control of viral infections. The authors summarize the proviral host factors and antiviral host factors in PEDV replication. This review is a logical and systematic summary of recent research on PEDV pathogens and their host interaction factors, and they also provide new insights into current issues related to the field of PEDV research.

The main issue is that we suggest that the authors provide a schematic diagram of the PEDV-host interaction mechanism to show the role and function of different host factors in PEDV replication.

Author Response

Dear editor and reviewers:

Thank you very much for your comments concerning our manuscript entitled “Porcine epidemic diarrhea virus: an updated overview of virus epidemiology, virulence variation patterns and virus-host interactions (viruses-1984092)”. Those comments are all valuable and very helpful for revising and improving our paper. We have revised the manuscript accordingly and a detailed response to the reviewers’ comments has been provided below.

To Reviewer #1:

Comments and Suggestions for Authors:

The PEDV, a member of the Coronaviridae family, causes acute diarrhoea and dehydration in pigs. In this manuscript, the authors summary the genotyping, distribution, origin, virion structure and function of PEDV. They also described the effects of PEDV viral proteins on virulence change and growth adaptation of strains from the perspective of gene heterogeneity. The key receptors of PEDV are still unknown. Therefore, the uncovering the infection mechanism by which PEDV invades host cells and achieves replication is important for the prevention and control of viral infections. The authors summarize the proviral host factors and antiviral host factors in PEDV replication. This review is a logical and systematic summary of recent research on PEDV pathogens and their host interaction factors, and they also provide new insights into current issues related to the field of PEDV research.

The main issue is that we suggest that the authors provide a schematic diagram of the PEDV-host interaction mechanism to show the role and function of different host factors in PEDV replication.

Response: Thanks very much for your advice. To better understand the biopathogenesis of PED, in the revised manuscript, we have mapped the network of PEDV-host interactions in Figure 3, and further elucidate the roles of PEDV components during viral infection. Furthermore, we have drawn a schematic diagram of PEDV infection in host cells and the role of pro/antiviral host factors in PEDV replication in Figure 4, which demonstrates the role and function of pro/antiviral host factors during PEDV infection.  The schematic is shown in the file.

Reviewer 2 Report

The author summarized the epidemiology and genetic evolution of PEDV, as well as the virus host interaction and antiviral related genes. The overall content of the article is detailed and clear, but there are the following problems:

1. The author needs to draw a Transmisson map of PEDV

2. Many mutations have occurred in PEDV during its epidemic evolution. Please discuss the trend of mutation changes during times

Author Response

Dear editor and reviewers:

Thank you very much for your comments concerning our manuscript entitled “Porcine epidemic diarrhea virus: an updated overview of virus epidemiology, virulence variation patterns and virus-host interactions (viruses-1984092)”. Those comments are all valuable and very helpful for revising and improving our paper. We have revised the manuscript accordingly and a detailed response to the reviewers’ comments has been provided below.

To Reviewer #2:

Comments and Suggestions for Authors:

The author summarized the epidemiology and genetic evolution of PEDV, as well as the virus host interaction and antiviral related genes. The overall content of the article is detailed and clear, but there are the following problems:

  1. The author needs to draw a Transmisson map of PEDV

Response: Thanks very much for your advice. In the revised manuscript, we have drawn a map of Transmisson of PEDV in Figure 1, which presents several major routes of transmission of PEDV. The schematic is shown in the file.

  1. Many mutations have occurred in PEDV during its epidemic evolution. Please discuss the trend of mutation changes during times

Response: Thank you very much for your helpful advice. In this revised manuscript, we have discussed the positions of amino acid mutations on PEDV S protein and the changes in amino acid biochemical properties, and found the trend of amino acid changes over time. The specific discussion content is shown below and marked in red:

In these PEDV strains, several potentially important amino acid mutations have been elucidated in a review[141]. Amino acid mutations with different properties accumulate in the virus strains with an increasing number of passages, while the mutational spectrum is stable over time, which may increase the stability of the virus in the host. The accumulation of amino acid substitutions is accompanied by drastic changes in amino acid properties, especially mutations in the S1 subunit of the S protein, with more than 93.3% of which resulting in transitions between hydrophilic and hydrophobic amino acids and changes in the charged properties of amino acids. However, the number of these transitions between hydrophilic and hydrophobic amino acids is essentially symmetrical in each strain, which may maintain the biochemical stability of viral proteins. As previously mentioned, the S1 subunit of the S protein is an important factor in determining PEDV virulence[124]. The PC220A-P120, PT-P96, YN144, FJzz1-F200, and OH851 strains were all mutated at domain 0 of the S1 subunit, which has the effect of binding sialic acid receptors on host cells to attach to virions[146]. Mutations in this domain may disrupt binding to sialic glycans and thereby reduce PEDV entry. The domains S A, S B, and S CD of the S1 subunit of the S protein have many B-cell epitopes that induce neutralizing antibodies[146,147], and mutations in these domains may affect the recognition of neutralizing antibodies. Especially in the CT-P120 and PT-P96 strains, the F636R and F636S mutations on the epitope COE are likely to reduce the reactivity of virus-neutralizing (VN) antibodies. The S2 subunit, which contains major immunodominant neutralizing epitopes[148], has a higher mutation frequency than the S1 subunit. More than 63% of mutations in the S protein occur in the S2 subunit. However, the mutations on the S2 subunit are milder. In YN144, PC22A-P120, and CT-P120 strains, 50-60% of the amino acid mutations did not change their biochemical properties, and the rest were changed in charge properties. While in PT-P96 and FJZZ1-F200 strains, there are still 85% of amino acid mutations with drastic changes in amino acid properties (charged, hydrophilic or hydrophobic), which may tolerate higher mutation pressure. Frequent mutations on the S2 subunit may be the main reason for the failure of neutralizing antibodies. Taken together, the attenuated virulence of these strains may be due to mutations in the S gene that impair the ability of receptor recognition and binding.

Reviewer 3 Report

In this review, the author summarizes the latest progress in the biology of PEDV. The global pandemic of PEDV result in a huge economic loss to the global swine industry. Author thoroughly described about PEDV structure and function, evolution, and virus-host interaction factors.  This is a well-written review which summarizes all the antiviral and proviral factors which may contribute to the pathogenesis of PEDV. It is clearly a big, concerted effort and well written review which sheds a new light on all the host factors affecting PEDV infection. Despite of all positive outcomes of the project, I found several points, where it can be improved.

Major comments –

1.     Author explained the virion structure of PEDV but did not explain the life cycle. It will be good if the author can show an illustration of life cycle also.

2.     In line 238 heading is Structural/accessory/non-structural proteins. It should be “variation in Structural/accessory/non-structural proteins”.

3.     Author shows all proviral and antiviral factors in a table. It will be good if the author can show it in illustration form. It will be more convincing and understandable to the readers.

Minor comment

1.     In line 245 margin is not according to the table. Please put it in the center or according to the table.

2.    In line 246 table 2 In the first column letter e is in the second line please put it in the same line

3.   Please reframe lines 286, 287, and 288.

4.  In line 32 please put a space between (PEDV) and [1] instead of (PEDV)[1]

5. In line 34 and most of the line you did not give a space before reference. Please give space before refence.

Author Response

Dear editor and reviewers:

Thank you very much for your comments concerning our manuscript entitled “Porcine epidemic diarrhea virus: an updated overview of virus epidemiology, virulence variation patterns and virus-host interactions (viruses-1984092)”. Those comments are all valuable and very helpful for revising and improving our paper. We have revised the manuscript accordingly and a detailed response to the reviewers’ comments has been provided below.

To Reviewer #3:

Comments and Suggestions for Authors:

In this review, the author summarizes the latest progress in the biology of PEDV. The global pandemic of PEDV result in a huge economic loss to the global swine industry. Author thoroughly described about PEDV structure and function, evolution, and virus-host interaction factors. This is a well-written review which summarizes all the antiviral and proviral factors which may contribute to the pathogenesis of PEDV. It is clearly a big, concerted effort and well written review which sheds a new light on all the host factors affecting PEDV infection. Despite of all positive outcomes of the project, I found several points, where it can be improved.

Major comments –

  1. Author explained the virion structure of PEDV but did not explain the life cycle. It will be good if the author can show an illustration of life cycle also.

      3.Author shows all proviral and antiviral factors in a table. It will be good if the author can show it in illustration form. It will be more convincing and understandable to the readers.

Response: Thanks very much for your advice. Combining the suggestions made in Comments 1 and 3, in the revised manuscript, we have drawn a schematic illustration of PEDV infection in host cells and the roles of pro/antiviral host factors in PEDV infection in Figure 4, which explains the life cycle of PEDV in infected host cells and function of pro/antiviral host factors in PEDV replication. The schematic is shown in the file.

  1. In line 238 heading is Structural/accessory/non-structural proteins. It should be “variation in Structural/accessory/non-structural proteins”.

Response: Thanks for your careful reading and thoughtful consideration. The heading “Structural/accessory/non-structural proteins” has been changed to “variation in Structural/accessory/non-structural proteins” in the revised manuscript.

Minor comment

  1. In line 245 margin is not according to the table. Please put it in the center or according to the table.

Response: Thanks for your careful review. We have corrected it in the revised manuscript.

  1. In line 246 table 2 In the first column letter e is in the second line please put it in the same line

Response: Thanks for your careful review. We have corrected it in the revised manuscript.

  1. Please reframe lines 286, 287, and 288.

Response: Thanks for your careful review. We have corrected it in the revised manuscript.

  1. In line 32 please put a space between (PEDV) and [1] instead of (PEDV)[1]
  2. In line 34 and most of the line you did not give a space before reference. Please give space before refence.

Response: Thanks for your careful review. We have corrected it in the revised manuscript.
